# Accumulated Knowledge of Activin Receptor-Like Kinase 2 (ALK2)/Activin A Receptor, Type 1 (ACVR1) as a Target for Human Disorders

**DOI:** 10.3390/biomedicines9070736

**Published:** 2021-06-26

**Authors:** Takenobu Katagiri, Sho Tsukamoto, Mai Kuratani

**Affiliations:** 1Research Center for Genomic Medicine, Division of Biomedical Sciences, Saitama Medical University, Hidaka-shi, Saitama 350-1241, Japan; stsukamo@saitama-med.ac.jp (S.T.); kuratani@saitama-med.ac.jp (M.K.); 2Project of Clinical and Basic Research for FOP, Saitama Medical University, Hidaka-shi, Saitama 350-1241, Japan

**Keywords:** activin receptor-like kinase 2 (ALK2), activin a receptor, type I (ACVR1), genetic disorder, animal model

## Abstract

Activin receptor-like kinase 2 (ALK2), also known as Activin A receptor type 1 (ACVR1), is a transmembrane kinase receptor for members of the transforming growth factor-β family. Wild-type ALK2/ACVR1 transduces osteogenic signaling in response to ligand binding. Fifteen years ago, a gain-of-function mutation in the *ALK2/ACVR1* gene was detected in patients with the genetic disorder fibro-dysplasia ossificans progressiva, which is characterized by heterotopic ossification in soft tissues. Additional disorders, such as diffuse intrinsic pontin glioma, diffuse idiopathic skeletal hyperostosis, primary focal hyperhidrosis, and congenital heart defects, have also been found to be associated with ALK2/ACVR1. These findings further expand in vitro and in vivo model system research and promote our understanding of the molecular mechanisms of the pathogenesis and development of novel therapeutics and diagnosis for disorders associated with ALK2/ACVR1. Through aggressive efforts, some of the disorders associated with ALK2/ACVR1 will be overcome in the near future.

## 1. Introduction

In 2006, a rare genetic disorder involving the skeletal muscle, called fibro-dysplasia ossificans progressiva (FOP; OMIM #135100), was reported to be associated with a recurrent mutation in the *Activin A receptor type I (ACVR1)* gene, which encodes the transmembrane receptor Activin receptor-like kinase 2 (ALK2)/ACVR1 protein [1] (Figure 1). FOP is a disorder characterized by heterotopic ossification in soft tissues, such as skeletal muscle, ligaments, and tendons [2,3]. Patients with FOP are subject to limited movement because heterotopic ossification fixes various joints. There were no effective treatments for heterotopic ossification in FOP [2,3]. The ALK2/ACVR1 protein serves as a receptor for members of the transforming growth factor-β (TGF-β) family, especially for members inducing osteogenic signaling, such as bone morphogenetic proteins (BMPs). This finding represents a turning point in basic, clinical, and translational research on ALK2/ACVR1 and its related disorders in the last 15 years. BMPs have been shown to be involved in the development of various organisms in vertebrates, such as bone, cartilage, brain, heart, lung, kidney, colon, and reproductive organs [4,5]. In addition to FOP, other disorders affecting the bone, brain, heart, and skin were also found to be associated with ALK2/ACVR1 [6,7,8,9,10,11,12,13] (Figure 1). Diffuse intrinsic pontine glioma (DIPG) is a sever pediatric brain tumor, and it is the second disorder that has been shown to be associated with ALK2/ACVR1 [6,7,8,9]. Similar to FOP, diffuse idiopathic skeletal hyperostosis (DISH; OMIM #106400) is a skeletal disorder which is characterized by the calcification and ossification of spinal ligaments and enthesis [10]. However, patients with DISH do not show heterotopic ossification, unlike patients with FOP. Congenital heart defects (CHDs) are commonly observed at birth, and patients with CHDs have abnormal structures of the heart and vessels that prevent normal blood flow [11,12]. Linkage analysis of patients with CHDs identified the ALK2/ACVR1 gene. Recently, primary focal hyperhidrosis (PFH) has also been shown to be associated with ALK2/ACVR1. Patients with PFH experience excessive sweating [13]. These findings further expand research on ALK2/ACVR1 and downstream signaling in in vitro and in vivo model systems, such as cell cultures and genetically engineered animals, respectively. Moreover, the findings allow for an understanding of the molecular mechanisms of the pathogenesis and development of novel therapeutics and diagnosis for disorders associated with ALK2/ACVR1.

## 2. Basics of Receptors of the TGF-β Family, Including ALK2/ACVR1

Among the TGF-β family, BMPs are well known to be important for skeletal development, including the development of bone and cartilage. Indeed, the unique activity of BMPs was found as a bone-inducing molecule in a demineralized bone matrix through an endochondral ossification process, which can be observed in long bones during embryonic development and fracture healing [14]. Since the bone-inducing activity was destroied by trypsin digestion, the molecule was named as a bone morphogenetic protein [15,16]. Experimental gene targeting and natural genetic disorders in molecules associated with BMP signaling, such as ligands and receptors, suggest that BMPs regulate the development of various organs in addition to the skeletal tissues, including teeth, skeletal muscle, tendons, ligaments, brain, eye, heart, lung, kidney, colon, and reproductive tissues. An aspect of this knowledge was found in relation to human disorders in ALK2/ACVR1.

### 2.1. Protein Structure of ALK2/ACVR1 in Vertebrates

Human ALK2/ACVR1 is a transmembrane protein that consists of 509 amino acids that can be divided into several domains: a signal peptide (amino acids 1–20), an extramembrane ligand-binding domain (amino acids 21–123), a transmembrane domain (amino acids 124–146), a glycine/serine (GS) domain (amino acids 178–207), and an intracellular kinase domain (amino acids 208–502) (Figure 2A). The human and mouse ALK2/ACVR1 proteins show 99.8% homology in overall amino acid sequences, though the signal peptides are slightly less homologous (85.0%) than the other domains (more than 95.5%). The structure of ALK2/ACVR1 proteins is highly conserved in vertebrates, from fish to humans (Figure 2B). However, alk8/acvr1 in *Danio rerio* consists of 506 amino acids and shows 65–68% homology with the proteins of other experimental animals; *Xenopus laevis* xAlk2/acvr1.S and *Gallus gallus* ALK2/ACVR1 have 507 and 504 amino acids, with 81–84% and 83–84% homology with others, respectively.

### 2.2. Type I and Type II Receptors of the TGF-β Family

In humans, twelve related transmembrane kinase proteins, including ALK2/ACVR1, function as binding receptors for TGF-β family ligands. They are classified into two subgroups, type I and type II receptors, based on the presence (type I) or absence (type II) of a characteristic stretch of glycine and serine residues (GS domain) in the cytoplasmic juxta-membrane domain [17,18]. Seven of the twelve receptors (ALK1 to ALK7) have a GS domain and are classified into type I receptors. The remaining five receptors (ActR-IIA, ActR-IIB, BMPR-II, AMHR-II, TbR-II) lack a GS domain and are classified as type II receptors [5,14,19].

Type I and type II receptors are also distinguished biochemically. The kinase domain of type I receptors is inactive without ligand binding; in contrast, type II receptor kinase domains are constitutively active. Bioactive ligands of the TGF-β family are dimeric proteins, with each monomer having two binding sites for type I and type II receptors [20,21,22]. Therefore, an active ligand is captured by two type I and two type II receptors on the cell membrane. In the ligand–receptor complex that is formed, active type II receptor kinases phosphorylate type I receptors in the cytoplasmic domain. Moreover, the GS domain of type I receptors has been shown to serve as a phosphorylation site by type II receptors [5,19]. Multiple serine and threonine residues in the GS domain are phosphorylated by type II receptors in response to ligand stimulation. The kinase activity of type I receptors is activated through phosphorylation of the GS domain by type II receptors, suggesting that phosphorylation changes the structural conformation of the GS domain. Accordingly, the GS domain acts as a “regulatory switch” for the enzyme activity of type I receptors.

### 2.3. The Role of Type I Receptors in Ligand-Induced Signal Transduction

Ligands of the TGF-β family have multiple biological activities under various conditions, including embryonic development, tissue maintenance and regeneration, and cell death. The ligands can be classified into osteogenic and non-osteogenic subgroups based on biological activity in a heterotopic bone formation assay in vivo [23,24]. BMP2, BMP4, BMP7, and BMP9 induce new bone formation in skeletal muscle tissues, whereas TGF-βs, Activin A, and myostatin do not exhibit such activity in vivo [23,24]. Osteogenic and non-osteogenic activities are determined via type I receptor binding, which activates intracellular signaling pathways. Osteogenic ligands bind to ALK1, ALK2, ALK3, and/or ALK6 as type I receptors and induce phosphorylation of the transcription factors Smad1 and Smad5; non-osteogenic ligands bind to ALK4, ALK5, and/or ALK7 and activate Smad2 and Smad3 [5,14]. In contrast to type I receptors, the binding capacities of type II receptors for ligands are broad, sharing both osteogenic and non-osteogenic ligands [25,26]. Therefore, for TGF-β family ligands, type I receptor kinases are the determinants of intracellular signaling and biological activities.

## 3. Disorders Associated with ALK2/ACVR1

Due to the importance of type I receptors in the TGF-β family, changes in structure by substitution mutations or ALK2/ACVR1 receptor expression levels are associated with several pathological conditions in humans.

### 3.1. Fibrodysplasia Ossificans Progressiva (FOP)

FOP is the first disorder shown to be associated with mutations of ALK2/ACVR1. Patients with FOP display abnormal new bone formation (heterotopic ossification) in skeletal muscle, tendons, and ligaments in childhood [2,3,27]. In addition, as acute heterotopic ossification is induced by trauma in those with FOP, invasive treatments are avoided. Most heterotopic ossification begins in childhood; it then spreads gradually throughout the body and fixes various joints, including the jaw, thus limiting range of motion [28,29]. Patients with typical FOP are distinguishable at birth from unaffected individuals due to malformations in the great toes, which are observed in more than 90% of patients with typical FOP [30,31,32]. The ossification process in FOP is similar to normal bone development and experimental bone induction by bone morphogenetic protein (BMP) in soft tissues. However, the heterotopic bone tissues formed in FOP are biochemically indistinguishable from those of the normal skeleton.

The incidence of FOP is estimated to be approximately 1 to 1.7–2 million neonates worldwide, regardless of race, sex, or region; FOP is transmitted in an autosomal dominant fashion and can be inherited from either the mother or father [2,3,29]. In 2006, five familial cases of FOP, with affected individuals showing both malformed toes and heterotopic ossification, were examined by genome-wide linkage analysis to identify the gene responsible for FOP; among these affected individuals, the heterozygotic single-nucleotide mutation c.617G>A was detected in the *ACVR1* gene, which is located on chromosome 2q [1]. Moreover, identical de novo mutations were observed in 32 of 32 sporadic cases of FOP [1]. *ACVR1* gene c.617G>A causes a substitution mutation of arginine to histidine at position 206 (p.R206H) of the ALK/ACVR1 protein, which is in the GS domain close to the juxta-membrane region (Figure 3). The ALK2/ACVR1 mutant p.R206H, but not wild-type, induces phosphorylation of Smad1 and Smad5, as well as BMP signaling in the absence of an exogenous ligand [33,34,35].

The p.R206H mutation has been found in more than 90% of patients with FOP. An additional 12 mutations, p.L196P, p.delP197_F198insL, p.R202I, p.Q207E, p.R258G, p.R258S, p.G325A, p.G328E, p.G328R, p.G328W, p. G356D, and p. R375P, have also been detected in patients with atypical FOP, with milder or more severe clinical features than typical p.R206H mutation-related FOP [36,37,38,39,40,41,42,43,44,45,46] (Figure 3). All mutations associated with FOP are in the intracellular, the GS, or the kinase domains [47] (Figure 3). FOP-associated ALK2/ACVR1 is further enhanced by activating BMP signaling via BMP type II receptors, such as ActR-IIB and BMPR-II [48,49], and such enhancement of mutant ALK2/ACVR1 depends on the kinase activity of type II receptors through the threonine at position 203 [48]. Kinase activity of type I receptors of the TGF-β family, including ALK2/ACVR1, is suppressed by FKBP12 [47,50,51]. One of the mutants associated with FOP, p.delP197_F198insL, involves a change at the FKBP12 binding site; indeed, the mutant was a registrant to FKBP12 in vitro [49]. Nevertheless, patients with FOP carrying the p.delP197_F198insL mutation exhibited equivalent or milder clinical features than those with typical FOP [38]. These findings suggest that a mechanism other than FKBP12 is involved in the activation of mutant ALK2/ACVR1 in patients with FOP.

Additionally, the p.R206H mutation renders the protein hypersensitive to BMPs [35]. Moreover, the mutant receptor becomes activated to induce BMP signaling through Smad1 and Smad5 by Activin A, which is a non-osteogenic ligand of the TGF-β family [52,53]. As ALK2/ACVR1 binds to Activin A, it is identified as “Activin A receptor, type 1”. Under normal conditions, Activin A binds to wild-type ALK2/ACVR1 and type II receptors as a nonsignaling complex at the cell membrane to block intracellular signaling activated by BMPs [53,54]. The change in biochemical function of ALK2/ACVR1 based on the ligand, such as Activin A, appears to be responsible for the pathogenesis of FOP.

Potential progenitor cells of chondrocytes and osteoblast in heterotopic ossification were examined using mice in vivo. In a BMP-induced heterotopic ossification assay in the skeletal muscle, both chondrocytes (Sox9+) and osteoblasts (Osterix+) were derived from Tie2+ cells, but not from endothelial cells (CD31+) or myoblasts (MyoD+) [55]. The Tie2+ cells were also PDGFRα+ and Sca-1+, suggesting that they are fibro/adipo-genic (FAP) cells in the skeletal muscle. The FAP cells express ALK2/ACVR1 and showed heterotopic ossification in mice carrying the p.R206H mutation [56]. Recently, the FAP cells were shown to express Activin A, a potential ligand for the induction of heterotopic ossification in FOP [57]. In addition to FAP cells, Scx+ cells were found to be involved in spontaneous ossification in tendons and ligaments in a mouse model carrying a p.R206H mutation in ALK2/ACVR1 [58]. Scleraxis (Scx) is a transcription factor specifically expressed in tenocytes and ligamental cells in vivo.

### 3.2. Diffuse Intrinsic Pontine Glioma (DIPG)

DIPG is a severe pediatric brain tumor without an effective treatment. In 2014, DIPG was the second example of a disorder associated with ALK2/ACVR1. More than 80% of cases of DIPG have been shown to be associated with mutations of histone H3.1 or H3.3. Whole-genome and whole-exome sequencing for patients with DIPG have revealed that some harbor additional somatic mutations of ALK2/ACVR1, including p.R206H, p.R258G, p.G328E, p. p.G328V, p.G328W, and p.G356D, similar to the germline mutations in patients with FOP [6,7,8,9] (Figure 3). Among them, only p.G328V is unique to DIPG; indeed, it is not found in other disorders, including FOP. Of the ALK2 mutants identified as related to DIPG and FOP, the p.G328V variant exhibits the highest kinase activity in cultured cells in vitro and in zebrafish in vivo [6,49]. BMP signaling regulates the differentiation of neural cells [59], and findings suggest that DIPG is caused by hyperactivation of BMP signaling through gain-of-function mutations in ALK2/ACVR1, similar to FOP. Nonetheless, DIPG may require additional mutations in histone H3.1 and H3.3, as well as in others. It has also been suggested that inhibitors against ALK2/ACVR1, such as small-molecule kinase inhibitors and/or downstream signaling, may be useful as novel therapeutics for both FOP and DIPG.

Using a mouse model carrying the ALK2ACVR1 p.G328V mutation, it was shown that the differentiation of oligo-dendro-gilial lineage cells was arrested, generating a high-grade diffuse glioma in a cooperation with Hist1h3b^K27M^ and Pik3ca^H1047R^.

### 3.3. Diffuse Idiopathic Skeletal Hyperostosis (DISH)

DISH is a skeletal disorder characterized by the ossification and calcification of spinal ligaments and entheses, which is observed more frequently in males than females. Furthermore, the incidence of DISH gradually increases in those aged over 70. Among potential genetic causes, the p.K400E mutation of ALK2/ACVR1 was identified in a patient with DISH in 2019 [10] (Figure 3).

Transient overexpression of ALK2/ACVR1 p.K400E in vitro resulted in hyperactivation of intracellular signaling by BMP2, BMP6, BMP7, and BMP9, but not non-osteogenic Activin A [60]. Moreover, the signaling-induction capacity of the p.K400E variant is enhanced by the presence of ActR-IIB, suggesting that this mutation is also a hypersensitive mutation against type II BMP receptors [60]. Although both DISH and FOP are skeletal disorders caused by the activation of ALK2/ACVR1, DISH does not involve heterotopic ossification of skeletal muscle tissues. The failure of the p.K400E variant to become activated by Activin A may explain the difference in clinical features between DISH and FOP.

It is unclear what type of cells express the mutant ALK2/ACVR1 in DISH. However, as they express ALK2/ACVR in mice, it is possible that ligamental cells are responsible for the pathogenesis of DISH [58].

### 3.4. Primary Focal Hyperhidrosis (PFH)

PFH is a potential genetic disorder characterized by excessive sweating, in part caused by sympathetic nervous system dysfunction. Recently, the gene responsible for PFH was mapped to human chromosome 2q31.1, which contains the *ACVR1* gene. It has been suggested that ALK2/ACVR1 is involved in the development of hair follicles. Expression of ALK2/ACVR1 can be detected in sweat gland tissues in both control and PFH groups, but the levels are higher in the former [13]. Overexpression of ALK2/ACVR1 in primary sweat gland cells causes an elevation in the expression levels of molecules involved in sweating, such as aquaporin 5 (AQP5) and sodium–potassium–chloride transporter 1 (NKCC1) [13].

### 3.5. Congenital Heart Defect (CHD)

BMPs are expressed in the heart during development and are implicated in atrioventricular septum development. Congenital heart defects (CHDs) are caused by multiple factors. Linkage and functional analyses of kindred have identified potential causative genetic mutations. Among 32 candidate genes from 190 patients, the *ALK2/ACVR1* gene was identified as being associated with CDHs in 2009 [11]. Three types of changes, namely p.A15G, p.R307L, and p.L343P, were observed in CHD, but only p.A15G was also found in 1.3% of controls. In 2011, p.H286N was detected in a patient with Down’s syndrome and CHD [12].

Overall, p.H286N shows weaker BMP activity than wild-type ALK2/ACVR1. Additionally, p L343P results in a decrease in protein levels and kinase activity in vitro [11,48]. In zebrafish, p.L343P of alk8/acvr1l acts as a dominant-negative BMP receptor [7]. However, co-expression of type II receptors, such as ActR-IIB and BMPR-II, fails to activate either p.H286N or p.L343P in vitro [48]. The pathological phenotypes in CHD suggest that cardiac myocytes and/or lineage cells may express ALK2/ACVR1 and affect development of the heart during embryogenesis. Taken together, CHDs are caused by reduced BMP signaling during development through a mutant receptor. This is in contrast to other disorders associated with ALK2/ACVR1, which are caused by the hyperactivation of downstream signaling.

The genetic mutations in ALK2/ACVR1 found in patients with FOP, DIPG, DISH, and CHD suggest a relationship between structure and function in signal transduction. Among the disorders, FOP, DIPG, and DISH are caused by gain-of-function of ALK2/ACVR1. The GS domain in type I receptors of the TGF-β family has been shown to act as a “regulatory switch” for the kinase in the receptors. The unphosphorylated GS domain masks the kinase to stabilize the inactive state of enzyme. The mutations in ALK2/ACVR1 found in FOP and DIPG, such as p.L196P, p.R202I, p.R206H, and p.R258S, p.G328E, p.G328R, p.G328W, p.G356D, and p.R375P, have been shown to be tightly clustered around the GS domain and the ATP-binding pocket in the kinase [47]. It was suggested that these mutations in ALK2/ACVR1 break the inhibitory interactions among the residues to stabilize the inactive state and shift toward an active state [47]. p.K400E, which was identified in DISH, was also shown to destabilize the ALK2/ACVR1 protein [10]. These findings in the disorders associated with gain-of-function mutations in ALK2/ACVR1 suggest an important role of intramolecular interactions to stabilize the kinase in inactive state.

## 4. Model Animals for Studying ALK2-Associated Disorders

### 4.1. Genomic Structure of Human and Mouse ACVR1 Genes

As mentioned above, human and mouse *ACVR1* show high similarity (Figure 4). Both are located on chromosome 2, and the protein coding region (from the initiation codon to the stop codon) consists of nine exons (Figure 4A). The 5′ untranslated region (UTR) is located in noncoding exons 1 and 2 and a part of coding exon 1; the 3′ UTR is at the 3′ end of coding exon 9. Each coding exon of *ACVR1* shows more than 85.1% homology between human and mouse, with identical numbers of nucleotides, whereas noncoding exons exhibit less than 40.3% homology (Figure 4B). Nine sets of primers have been designed to amplify and sequence the nine coding exons for the diagnosis of human disorders associated with mutations in the ALK2/ACVR1 protein, and these primers were optimized for use at once under a single common condition [61].

### 4.2. Genetically Engineered Animal Models of ALK2/ACVR1-Related Disorder

The extracellular (signal peptide and ligand binding domain) and intracellular domains (GS and kinase domain) of ALK2 are encoded by exons 1 to 3 and exons 3 to 9 in both humans and mice, respectively (Figure 5A). Several animal models have been developed by engineering the *ACVR1* gene to examine the function of ALK2/ACVR1 in vivo (Figure 5B).

To dissect the physiological function of ALK2/ACVR1, a universal ALK2/ACVR1-deficient mouse line was established by removing coding exon 4, which contains the GS domain, in mouse ES cells [62]. ALK2/ACVR1 deficiency is lethal in early embryonic development by E9.5. A Cre-LoxP system was also applied to generate conditional *ACVR1*-knockout mice. Two LoxP sites were inserted into the introns 5′ and 3′ of coding exon 6 [63], and transient infection of the modified ES cells with adenovirus expressing Cre led to recombination between the LoxP sites, resulting in a lack of the target exon. The developed floxed mice can be mated with other mice expressing tissue-specific Cre to establish tissue-specific conditional knockout mice. To target the GS domain, another floxed mouse line was established to remove coding exon 4 [64].

Gain-of-function models of ALK2/ACVR1 have also been developed in mice and zebrafish via exogenous transgene expression. Substitution of a glutamine residue at position 207 to aspartic acid (p.Q207D), which was modified in exon 4, constitutively activates the kinase activity of human and mouse ALK2/ACVR1. The active mutant Alk2/Acvr1 cDNA was placed downstream of LoxP sites and the triple poly A signal for conditional expression under Cre-dependent recombination in vivo. Mice injected locally with Cre-expressing adenovirus show heterotopic ossification [65]. Glutamine residue 207 in human and mouse ALK2/ACVR1 corresponds to glutamine residue 204 in zebrafish alk8/acvr1l. Transgenic zebrafish were also developed by expressing a p.Q204D transgene [66]. In contrast to the mouse model, zebrafish carrying the p.Q204D variant did not develop heterotopic ossification after injury [67]. The *Alk2/Acvr1* gene was engineered to change in p.R206H in mouse ES cells to produce *Acvr1^R206H/+^* as a mouse model of FOP [68]. Chimeric mice carrying the mutant ES cells show heterotopic ossification and malformation of the great toe, similar to patients with FOP; however, mouse lines cannot be generated [68]. A conditional knock-in mouse line carrying p.R206H was developed to generate an inducible p.R206H variant. In this model, wild-type human exon 4 and mutated mouse exon 4 were placed with two sets of LoxP- and LoxP-related sites in the mouse *Alk2/Acvr1* gene [53]. After recombination by the Cre enzyme, the cells express p.R206H instead of wild-type Alk2/Acvr1. In this model, an anti-Activin A blocking antibody was able to inhibit heterotopic ossification in vivo [53]. Another conditional knock-in mouse line carrying p.R206H was developed to examine heterotopic ossification in vivo. The cassette of CAG-tdTomato-stop was placed in the intron with the R206H mutation in exon 4, and the mice expressed p.R206H and GFP to identify cells undergoing Cre-dependent recombination [56]. Heterotopic ossification in these mice was increased by removing the wild-type allele of Alk2/ACVR1, suggesting that the wild-type receptor acts as a suppressor against mutant Alk2/Acvr1 [56]. Moreover, the ALK2/ACVR1 p.G328V variant, a mutation in coding exon 6, is a unique mutation in DIPG. A conditional knock-in p.G328V mouse line was established by the insertion of a transcriptional termination sequence and two LoxP sites into the intron 5′ of coding exon 7 with the p.G328V mutation [69]. The expression of p.G328V in mouse brain blocked oligodendrocyte differentiation. Moreover, the mutant mice displayed high-grade glioma in combination with mutations observed in patients with DIPG, such as *Hist1h3b^K27M^* and *Pic3ca^H1047^*.

The animal models established by modifying ALK2/ACVR1 have been used for the development of new effective treatments and drugs for ALK2/ACVR1-associated disorders. Due to the fact that FOP and DIPG are caused by gain-of-function mutations in ALK2/ACVR1, various types of inhibitors have been developed to suppress excess intracellular signaling. Dorsomorphin and LDN-193189, which are small chemical inhibitors of ALK2/ACVR1 kinase, inhibit BMP activity in vitro and in vivo through p.R206H of ALK2/ACVR1 [35,65]. LDN-212854 has been developed as a specific kinase inhibitor for ALK2/ACVR1 [70]. Saracatinib (AZD0530), which is one of the kinase inhibitors approved by the FDA, was found to inhibit ALK2/ACVR1 [71]. RARγ agonists suppressed chondrogenesis in vitro through decreasing transcription factor Smads, and inhibited heterotopic ossification induced by BMP2 in vivo [72]. Palovarotene, a RARγ agonist, was applied to inhibit heterotopic ossification in mice and patients with FOP [73], (Clinicaltrials.gov registration NCT03312634). Rapamycin, also known as sirolimus, is an immunosuppressor and was found to be an effective inhibitor of chondrogenesis in FOP patient-derived iPS cells [74]. A blocking antibody against Activin A was also shown to inhibit heterotopic ossification in a mouse model of FOP [53]. Many types of additional inhibitor for ALK2/ACVR1 signaling are currently being examined in preclinical studies.

## 5. Conclusions

ALK2/ACVR1 is a type I receptor for TGF-β family members, especially osteogenic ligands that activate intracellular signaling through Smad1 and Smad5. Over the last 15 years, the identification of a direct association between ALK2/ACVR1 and human disorders, including FOP, DIPG, DISH, CHD, and PFH, has led to extensive research in basic, clinical, and translational medicine concerning this receptor, and various in vitro and in vivo experimental models have been developed to understand the functions of ALK2/ACVR1 and to reveal the molecular mechanisms of the pathogenesis of related disorders. These studies have shown that FOP, DIPG, DISH, and PFH may be caused by ALK2/ACVR1 receptors with gain-of-function mutations. Therefore, inhibitors of intracellular signaling through ALK2/ACVR1 would constitute novel therapeutics for related disorders. This possibility has been studied using experimental models of ALK2/ACVR1 in vitro and in vivo. Some of the disorders associated with ALK2/ACVR1 will be overcome in the near future through aggressive efforts.

## Figures and Tables

**Figure 1 biomedicines-09-00736-f001:**
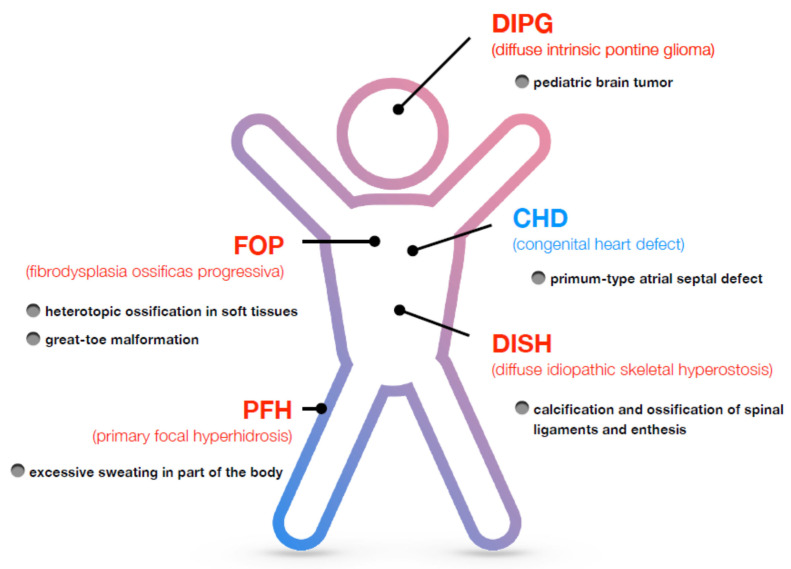
Disorders associated with ALK2/ACVR1. Five types of disorders reported to be associated with ALK2/ACVR1 are summarized, along with their major clinical features. Fibro-dysplasia ossificans progressiva (FOP), diffuse intrinsic pontine glioma (DIPG), diffuse idiopathic skeletal hyperostosis (DISH), and primary focal hyperhidrosis (PFH) may be caused by overactivation of ALK2/ACVR1. In contrast, congenital heart defects (CHDs) are caused by suppression of ALK2/ACVR1 signaling. Please see details in the text.

**Figure 2 biomedicines-09-00736-f002:**
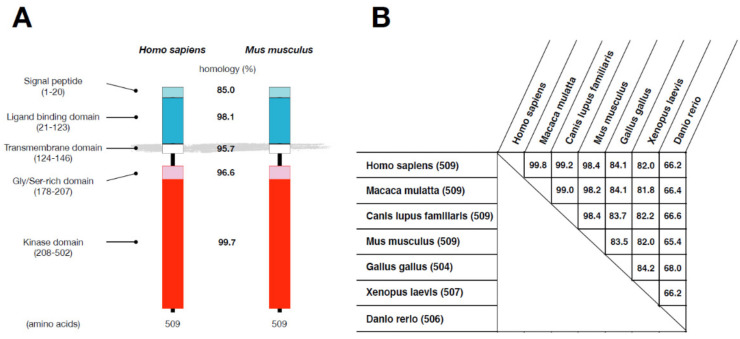
ALK2/ACVR1 in different species: (**A**) Human and mouse ALK2/ACVR1. The amino acid-based homology of each domain was calculated by Genetyx-Mac version 20 (Genetyx Co., Tokyo, Japan). (**B**) Homology of ALK2/ACVR1 among animal species. UniProt ID of animal species (Homo sapiens: Q04771, Macaca mulatta: F7A9J8, Canis lupus familiaris: A0A5F4C195, Mus musculus: P371728, Gallus gallus Q90ZK6, Xenopus laevis O42475, Danio rerio: Q9DGI6). Complete amino acid sequences of ALK2/ACVR1 were compared using Genetyx-Mac version 20.

**Figure 3 biomedicines-09-00736-f003:**
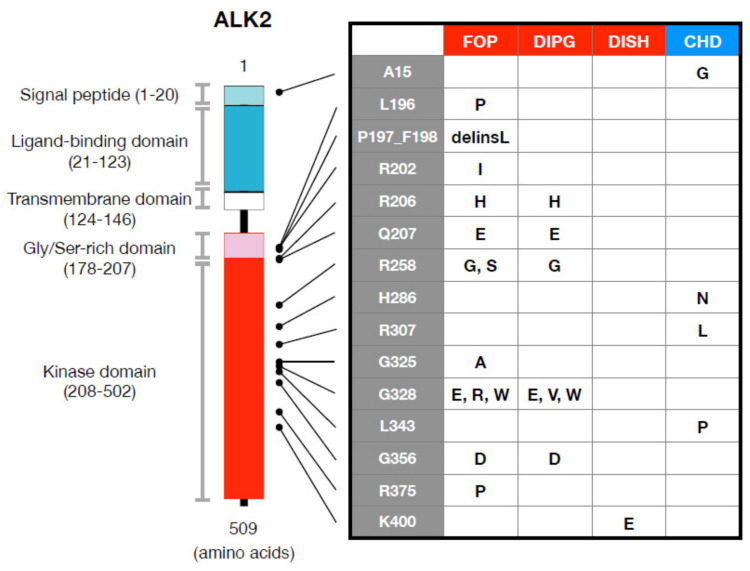
Basic structure of ALK2/ACVR1 and mutations reported in genetic disorders. Human ALK2 consists of 509 amino acids: 1–20, signal peptides; 21–123, ligand-binding domain in the extracellular space; 124–146, transmembrane domain; 178–207, glycine (Gly) and serine (Ser)-rich domain, which is a regulatory switch of the kinase; 208–502, a serine and threonine kinase domain, which phosphorylates downstream substrates in cytoplasm. Amino acids with positions in wild-type ALK2/ACVR1 (the first row from left in a table) are substituted in fibro-dysplasia ossificans progressiva (FOP), diffuse intrinsic pontine glioma (DIPG), diffuse idiopathic skeletal hyperostosis (DISH), and primary focal hyperhidrosis (PFH). Please note that several substitutions overlap in FOP and DIPG. Capital letters in the table are amino acid residues mutated in the disorders.

**Figure 4 biomedicines-09-00736-f004:**
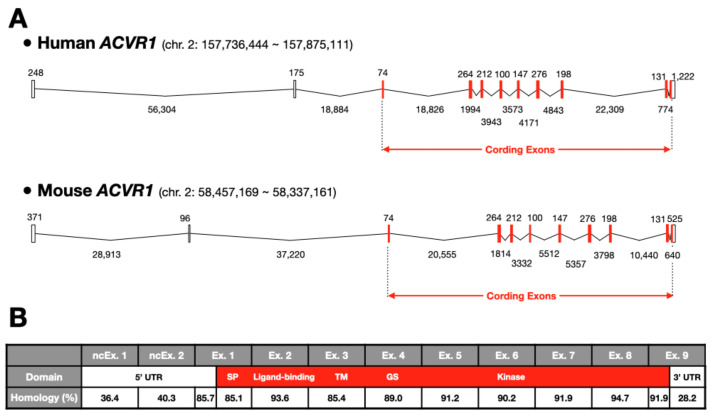
Genomic structures of human and mouse ALK2/ACVR1: (**A**) Genomic structures of humans: ENST00000263640.7 (upper) and mice: ENSMUST00000056376.12 (lower) ALK2/ACVR1 on each chromosome 2 (chr.) are schematically indicated. Open and closed boxes indicate noncoding (nc) and coding exons, respectively. Lines are introns. Numbers are the length of nucleotides. (**B**) Homology of each exon between human and mouse ALK2/ACVR1 genes. Encoded domains are the signal peptide (SP), ligand-binding, transmembrane (TM), glycine/serine domain (GS), and kinase domain. UTR—untranslated region. Please note that the homology of coding exons is higher than that of noncoding exons.

**Figure 5 biomedicines-09-00736-f005:**
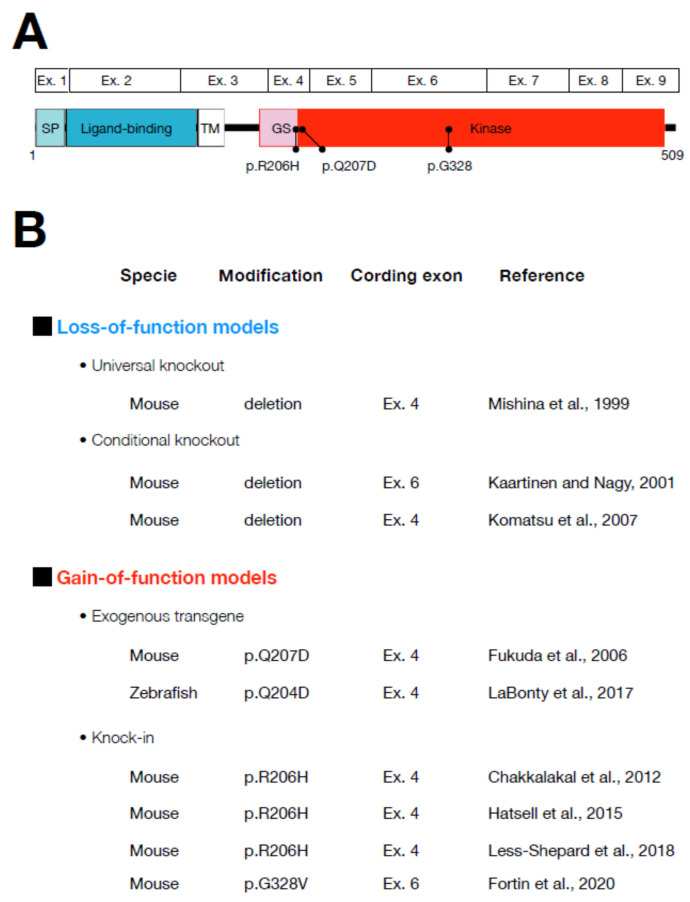
Animal models established by modifying the ALK2/ACVR1 gene: (**A**) Schematic representation of the ALK2/ACVR1 protein and coding exons (Ex.). (**B**) Animal models in mice and zebrafish have been established to study the physiological functions and pathological changes of ALK2/ACVR1. They are classified in loss-of-function and gain-of-function models. Animal species, modifications, modified target coding exons, and references are summarized.

## Data Availability

Not applicable.

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
