# Peer review of "Accumulated Knowledge of Activin Receptor-Like Kinase 2 (ALK2)/Activin A Receptor, Type 1 (ACVR1) as a Target for Human Disorders"

_biomedicines, 2021, doi:10.3390/biomedicines9070736_

Round 1
Reviewer 1 Report
In this review, the authors described that the involvement of ALK2/ACVR1 receptors in fibrodysplasia ossificans progressiva (FOP), diffuse intrinsic pontine glioma (DIPG), diffuse idiopathic skeletal hyperostosis (DISH), primary focal hyperhidrosis (PFH), and congenital heart defect (CHD). However, this review does not bring important information to the potential readers because they did not describe the mechanism how ALK2/ACVR1 receptors contribute the induction of these diseases. The following comments may help in revising manuscript.
1) What cells are responsible for the disorder(s) of ALK2/ACVR1 receptors in each diseases?
Are the disorders of found in fibroblasts or osteoblasts in case of fibrodysplasia ossificans progressiva (FOP)? Are the disorders of found in glioma cell(s) in case of diffuse intrinsic pontine glioma (DIPG)?
The authors should describe what cells are responsible for the disorder(s) of ALK2/ACVR1 receptors in each diseases so that the potential readers would understand the disorder(s) of ALK2/ACVR1 receptors were found in different cells type of cells.
2) Different of function(s) of ALK2/ACVR1 receptors in different type of cells.
The authors described that involvement in FOP, DIPG, DISH, and PFH. Because the mechanism to induce these diseases would be totally different, the function of ALK2/ACVR1 receptors would be different. It is not surprising the function of ALK2/ACVR1 receptors would be different if the disorders are found in different type of cells.
The authors should describe how the disorders of ALK2/ACVR1 receptors in each cell contribute the induction of FOP, DIPG, DISH, and PFH.
3) Is it possible that ALK2/ACVR1 receptors is the target for the treatment of FOP, DIPG, DISH, and PFH ?
Most of the potential readers wants to know whether ALK2/ACVR1 receptors is potential target for the treatment of FOP, DIPG, DISH, and PFH.
Although the authors described that FOP, DIPG, DISH, and PFH may be caused by ALK2/ACVR1 receptors with gain-of-function mutations, they did not describe ALK2/ACVR1 receptors the primary target of these diseases.
Reviewer 2 Report
The review entitled: “Accumulated Knowledge of Activin Receptor-Like Kinase 2(ALK2)/Activin A Receptor, Type 1 (ACVR1) as a Target for Human Disorders” by Katagiri et all summarizes the literature on the role of Activin A receptor type 1 (ACVR1) as a transmembrane kinase receptor for members of the transforming growth factor-b family in subset of human disorders.
The topic is of high interest as it deals with rare genetic mutations and severe pathologies. Apparently, we all agree that the more we know about the genetic factors involved in rare diseases, the more precise therapies will be developed. The last is a prerequisite for
My overall rate of the work is good and I suggest publication in the journal after certain amendments, which I advise the authors to make.
Below, please find attached my comments:
MAJOR
- The Introduction part starts with a sentence which is not applicable: @Line 25: “The identification of a gene responsible for a genetic disorder frequently sheds light on unknown functions of a protein and unveils molecular mechanisms of a biological ”. I find it inappropriate because the beginning sounds like a introduction of a schoolbook, rather than a scientific article. I advise the authors to edit this sentence and start with the gene they discuss later in their work.
- The Introduction part is shorter than accepted. I advise the authors to include some medical and clinical data for the diseases and pathologies the gene ALK2/ACVR1 is involved in. The mere mentioning of them is insufficient for the reader to understand the importance of the cases, nor the importance of the role of this particular gene in them.
- The protein structure and interspecies homology is important, though the authors need to shed some light on the importance of the chemical structure of the protein encoded by the gene of interest and moreover to shed light on the role of mutations in that protein and gene structure for the organism.
- Moreover, I would advise the authors to include some data for the known roles of the transforming growth factor-b (TGF-b) family in the organismal development. This will strengthen the importance of the review.
- The animal models for studying mutations in ALK2/ACVR1 are well-described. What I find necessary is a detailed explanation for what exactly therapeutic approaches these models are used as the main conclusion of the review is: “Therefore, inhibitors of intracellular signaling through ALK2/ACVR1 would constitute novel therapeutics for disorders. This possibility has been studied using experimental models of ALK2/ACVR1 in vitro and in vivo. Some of the disorders associated with ALK2/ACVR1 will be overcome in the near future through aggressive efforts.” I definitely need some data in the review on these inhibitors and their role in treating the reported disorders. Information for this we see only @Line 182 regarding DIPG. I advise the authors to broaden their review and provide more detailed data on these inhibitors and not only for this pathology but for the others discussed in the review and influenced by the gene of interest.
MINOR
- There are some typos and grammatical errors, please revise and edit.
- “in vivo” and “in vitro” have to be in Italic, I advise you to check the whole manuscript and edit where necessary.
Round 2
Reviewer 1 Report
The authors responded all the concerns appropriately.